# MAXIMUM POWER OF SALINE AND FRESH WATER MIXING IN ESTUARIES

Zhilin Zhang[1,2] and Hubert Savenije[1]

[1]Department of Water Management, Delft University of Technology, Delft, the Netherlands.
[2]Guangdong Research Institute of Water Resources and Hydropower, Guangzhou, China.

**Correspondence:** Hubert Savenije (h.h.g.savenije@tudelft.nl)

**Abstract.** According to *Kleidon* (2016), natural systems evolve towards a state of maximum power, leading to higher levels of entropy production by different mechanisms, including the gravitational circulation in alluvial estuaries. Gravitational circulation is driven by the potential energy of the fresh water. Due to the density difference between seawater and riverwater, the water level on the river side is higher. The hydrostatic forces on both sides are equal, but have different lines of action. This triggers an angular moment, providing rotational kinetic energy to the system, part of which drives mixing by gravitational circulation, lifting up heavier saline water from the bottom and pushing down relatively fresh water from the surface against gravity; the remainder is dissipated by friction while mixing. With a constant freshwater discharge over a tidal cycle, it is assumed that the gravitational circulation in the estuarine system performs work at maximum power. This rotational flow causes the spread of salinity inland, which is mathematically represented by the dispersion coefficient. In this paper, a new equation is derived for the dispersion coefficient related to density-driven mixing, also called gravitational circulation. Together with the steady state advection-dispersion equation, this results in a new analytical model for density-driven salinity intrusion. The simulated longitudinal salinity profiles have been confronted with observations in a myriad of estuaries worldwide. It shows that the performance is promising in eighteen out of twenty-three estuaries that have relatively large convergence length. Finally, a predictive equation is presented to estimate the dispersion coefficient at the downstream boundary. Overall, the maximum power concept has provided a new physically based alternative for existing empirical descriptions of the dispersion coefficient for gravitational circulation in alluvial estuaries.

## 1 Introduction

Estuaries are water bodies where rivers with fresh water meet the open sea. The longitudinal salinity difference causes a water level gradient along the estuary. As a result, the water level at the limit of salt intrusion is set-up several cm above sea level (about 0.012 times the estuary depth). Therefore, the hydrostatic forces from seaside and riverside have different lines of action (a third of the set-up apart). Since the hydrostatic forces at the sea side and the salinity limit are equal but opposed, this

difference in the lines of action triggers an angular moment (a torque) which drives the gravitational circulation, whereby fresh water near the surface flows to the sea and saline water near the bottom moves upstream (*Savenije*, 2005). This density-driven gravitational circulation is one of the two most significant mixing mechanisms in alluvial estuaries; the other is the tide-driven mixing mechanism which can be dominant in the wider (downstream) part of estuaries (*Fischer et al.*, 1979).

*Kleidon* (2016) described the concept of maximum power in the Earth system, implying that freely evolving systems perform work and dissipate energy at maximum power (close to or at the Carnot limit). Using this concept, gravitational circulation is assumed to take place at the maximum power limit. Earlier, the maximum power concept was used to solve the saline and fresh water mixing as in a thermodynamic equilibrium system (*Zhang and Savenije*, 2018). It assumed that in thermodynamic terms, the freshwater flux maintains a potential energy gradient, triggering fresh and saline water mixing processes that work

at depleting this gradient. Because the strength of the mixing in turn depends on this gradient, there is an optimum at which the mixing process performs at maximum power. It did, however not account for the energy loss associated with this mixing process. The equation obtained appeared to have an analytical solution of a straight line for the longitudinal salinity distribution. Although this is not correct, it can be seen as a first order approximation, which agrees with earlier theoretical work by *Hansen and Rattray* (1965), who developed their theory for the central region of the salt intrusion length where the salinity gradient

is at its maximum and dominated by density-driven mixing. However, this approximate solution was not fully satisfactory for simulating the salinity distribution.

     In contrast to the earlier work by *Zhang and Savenije* (2018), in this paper friction is taken into account. The available free energy by the angular moment, is converted into work (mixing the saline and fresh water against the force of gravity) and the associated frictional dissipation. In the following sections we shall derive a new equation for density driven mixing, which

appears to compare well with observations in a range of alluvial estuaries.

     *Kleidon* (2016) presented several examples for the application of the maximum power limit on non-thermal energy conversions. In one example, a fluid is kept in motion by an accelerating force which provides kinetic energy to the system. The velocity of the fluid is slowed down by friction and the remainder is converted into another form of energy. If the velocity is too large, the friction is large and energy dissipation dominates, then the power of the force to generate work is limited. In contrast,

if the velocity is too small, the power is not enough to generate work. Hence, there is an optimum value for the product of the force and velocity to produce maximum useful energy. Estuaries are comparable to this system. In this article, we apply the maximum power concept to gravitation circulation generated by a longitudinal density gradient.

     Traditionally, the empirical Van der Burgh (VDB) method has worked very well to describe the mixing in alluvial estuaries leading to predictive equations to describe the salinity intrusion in alluvial estuaries (*Savenije*, 2005, 2012). The VDB method

takes account of all mixing mechanisms, including density-driven (gravitational) circulation and tide-driven mixing. For application of the VDB method, there are two parameters that need to be calibrated, the empirical Van der Burgh coefficient $K$ and the dispersion coefficient at the downstream boundary $D_0$. This method has performed surprisingly well around the world and has been used in this paper as the benchmark model for comparison with the maximum power approach.

## 2  Moment balance for an open estuary system

In an estuary, the cross-sectional average hydrostatic forces have equal values along the estuary axis. Over a segment, the forces are opposed but working on different lines of action due to the density gradient in upstream and downstream directions. As a result, they exert an angular moment (torque) $M_{\mathrm{acc}}$ that drives the gravitational circulation, performing as accelerating torque. The velocity of the gravitational circulation kept in motion by this accelerating torque is slowed down by a friction moment $M_{\mathrm{fric}}$, which is the product of the associated friction force and its arm. The remainder $M_{\mathrm{ex}}$ drives the circulation and execute work against gravity (Figure 1). Hence, the balance in steady state in a segment is

$$M_{\mathrm{acc}} - M_{\mathrm{ex}} - M_{\mathrm{fric}} = 0 \ . \tag{1}$$

The moment due to the friction against the circulation is expressed as

$$M_{\mathrm{fric}} = F_{\mathrm{fric}} l_m \ , \tag{2}$$

with $F_{\mathrm{fric}}$ being the friction force in N and $l_m$ the scale of the arm of the frictional forces in m.

The friction force during the dispersive circulation is expressed as

$$F_{\mathrm{fric}} = \tau O \ , \tag{3}$$

where $\tau$ is the shear stress in $\mathrm{Nm}^{-2}$ and $O$ is the contact area in $\mathrm{m}^2$. Estuarine mixing has two length scales: a vertical and a horizontal one. The horizontal length scale is the tidal excursion $E$, which is the distance a water particle travels on the tide; the vertical length scale is the depth $h$, over which saline water is moved upward to the surface, and over which relatively fresh water is moved downward to the bottom. Since the process of gravitational mixing is essentially to move the saline water up and the fresher water down, the contact area for the resistance against this movement is determined by the depth ($h$) and the width ($B$). Following that reasoning, $O$ is assumed equal to $Bh$. Meanwhile, the circulation cell has a dimension constrained by the depth. The circular movement hence has a diameter of the depth and $l_m$, the horizontal arm between the vertical frictional forces, is in the order of magnitude of the depth.

### 2.1  Maximum power condition in estuaries

Because the velocity of the dispersive gravitational circulation is small, the mixing flow is assumed to be laminar. The shear stress is typically a function of flow velocity ($v$): $\tau = \rho q v$, with $\rho$ being the density in $\mathrm{kgm}^{-3}$ and $q$ being a laminar resistance in $\mathrm{ms}^{-1}$. The latter is assumed to be proportional to the tidal velocity amplitude ($q \propto E/T$), where $T$ is the tidal period in s. Hence, the flow velocity representing the gravitational circulation is:

$$v = \frac{M_{\mathrm{acc}} - M_{\mathrm{ex}}}{\rho q B h^2} \ . \tag{4}$$

Power is defined by the product of a force and its velocity. The power of torque (angular moment) is defined as the product of the moment and its angular velocity. Hence, the power is defined as

$$P = M_{\text{ex}}\omega = M_{\text{ex}}\frac{v}{h/2}2\pi = \frac{\pi}{\rho q B h^3}\left(M_{\text{acc}} - M_{\text{ex}}\right)M_{\text{ex}} , \tag{5}$$

where $\omega$ is the angular velocity or the rotational speed in $\text{s}^{-1}$. Figure 2 illustrates how the execution moment and the flow
velocity vary. If the working moment is too large and causes fast mixing flow, the energy dissipation is large and diminishes the flow velocity. If it is too small, the mixing would also be sub-optimal. In analogy with *Kleidon* (2016), the product of the working moment and the flow velocity has a well-defined maximum. The maximum power (MP) is then obtained by: $\partial P/\partial M_{\text{ex}} = 0$. Hence, the optimum values of the execution moment $M_{\text{ex,opt}}$ and the flow velocity $v_{\text{opt}}$ are

$$M_{\text{ex,opt}} = \frac{1}{2}M_{\text{acc}} \tag{6}$$

and

$$v_{\text{opt}} = \frac{M_{\text{acc}}}{2\rho q B h^2} . \tag{7}$$

Here, the accelerating force ($F_{\text{acc}}$) that produces the angular moment is the hydrostatic force obtained by integrating the hydraulic pressure over the depth:

$$F_{\text{acc}} = \frac{1}{2}\rho_0 g h^2 B , \tag{8}$$

where $\rho_0$ is the density of the seaside in $\text{kgm}^{-3}$.

The accelerating moment has an arm $\Delta h/3$ (*Savenije*, 2005). The water level gradient according to the balance of the hydrostatic pressures results in:

$$\frac{\mathrm{d}h}{\mathrm{d}x} = -\frac{h}{2\rho}\frac{\mathrm{d}\rho}{\mathrm{d}x} , \tag{9}$$

where $x$ is the distance in m. Density is a function of salinity ($S$ in psu): $\rho = \rho_f(1 + c_S S)$, where $\rho_f$ is the density of the
freshwater in $\text{kgm}^{-3}$ and $c_S$ ($\approx 7.8 \times 10^{-4}$) is the saline expansivity in $\text{psu}^{-1}$.

Subsequently, the accelerating moment due to the density gradient driving gravitational circulation over a tidal cycle can be described as:

$$M_{\text{acc}} = F_{\text{acc}}\frac{1}{3}\frac{\mathrm{d}h}{\mathrm{d}x}E = -\frac{1}{12}\rho_0 g h^3 B c_S \frac{\mathrm{d}S}{\mathrm{d}x}E , \tag{10}$$

where $E$ is the horizontal length scale of the gravitational circulation in m.
In steady state, the one-dimensional advection-dispersion equation averaged over the cross section and over a tidal cycle reads (*Savenije*, 2005, 2012):

$$|Q|S + AD\frac{\mathrm{d}S}{\mathrm{d}x} = 0 , \tag{11}$$

where $Q$ is the freshwater discharge in $\mathrm{m^3\,s^{-1}}$, $A\,(=Bh)$ is the cross-sectional area in $\mathrm{m^2}$, and $D$ is the dispersion coefficient in $\mathrm{m^2\,s^{-1}}$. The positive direction of flow is in the upstream direction.

Accordingly, with $q \propto E/T$, the optimum velocity is

$$v_{\mathrm{opt}} \propto \frac{c_S g h T}{24} \frac{|Q|S}{AD} \ . \tag{12}$$

Assuming that the steady state over a tidal cycle is driven mainly by the accelerating moment especially in the upstream part where tidal mixing is relatively small and this gravitational circulation ($D_g$) is proportional to the dispersive residual velocity ($D_g \propto v_{\mathrm{opt}} E$),

$$D_g \propto \left( \frac{c_S g}{24} \frac{S|Q|ET}{B} \right)^{1/2} \ . \tag{13}$$

This equation indicates that the dimensionless dispersion coefficient is proportional to the root of the estuarine Richardson number $N_R$:

$$\frac{D_g}{\upsilon E} \propto N_R^{0.5} = \left( c_S S \frac{gh}{\upsilon^2} \frac{|Q|T}{AE} \right)^{0.5} , \tag{14}$$

where $\upsilon$ is the tidal velocity amplitude in $\mathrm{m\,s^{-1}}$. The Richardson number describes the balance between the potential energy of the fresh water flowing into the estuary ($\Delta\rho g h |Q|T/2$) and the kinetic energy of the tidal flood flow ($\rho\upsilon^2 AE/2$) (*Fischer et al.*, 1979; *Savenije*, 2005; *Zhang and Savenije*, 2017).

## 2.2 Analytical solution for the dispersion equation

Equations derived from the maximum power concept are obtained along the estuary axis, hence they can be used at any segment along the estuary. Then, equation (13) becomes

$$D_g(x) = C_3 \left( \frac{S|Q|ET}{B} \right)^{1/2} , \tag{15}$$

where $C_3$ is a factor in $\mathrm{psu^{-1}ms^{-2}}$ and all local variables are a function of $x$.

The following equations are used for the tidal excursion and width in alluvial estuaries:

$$E(x) = E_0 e^{\delta_H(x-x_0)} , \tag{16}$$

$$B(x) = B_0 e^{-(x-x_0)/b} , \tag{17}$$

where $\delta_H$ is the tidal damping rate in $\mathrm{m^{-1}}$ and $b$ is the geometric convergence length of the width in m. A smaller $b$ value implies stronger convergence (a stronger funnel shape). The subscript '0' represents parameters at the geometric boundary condition ($x = x_0$).

At the boundary, equation (15) is given by:

$$D_{g0} = C_3 \left( \frac{S_0 |Q| E_0 T}{B_0} \right)^{1/2} . \tag{18}$$

Substitution of equations (16)–(18) into (15) gives

$$D_g(x) = D_{g0} \left( \frac{S}{S_0} \right)^{1/2} e^{\Omega(x-x_0)} , \tag{19}$$

with $\Omega = \delta_H/2 + 1/(2b)$.

Differentiating $D_g$ with respect to $x$ and using the steady state equation (11) results in

$$\frac{\mathrm{d}D_g}{\mathrm{d}x} = \frac{D_g}{2S} \frac{\mathrm{d}S}{\mathrm{d}x} + \Omega D_g = \Omega D_g - \frac{1}{2} \frac{|Q|}{A} . \tag{20}$$

The cross-sectional area $A$ is given by

$$A(x) = A_0 e^{-(x-x_0)/a} , \tag{21}$$

where $a$ is the convergence length of the cross-sectional area in m.

Substituting equation (21) into (20) and in analogy with *Kuijper and Van Rijn* (2011) and *Zhang and Savenije* (2017), the solution of the linear differential equation (20) is

$$\frac{D_g}{D_{g0}} = e^{\Omega(x-x_0)} - \frac{|Q|\zeta}{2A_0 D_{g0}} \left[ e^{(x-x_0)/a} - e^{\Omega(x-x_0)} \right] , \tag{22}$$

with $\zeta = a/(1 - \Omega a)$.

At the salinity intrusion limit ($x = L$), $D_g = 0$, resulting in

$$L = \zeta \ln \left( 1 + \frac{2A_0 D_{g0}}{|Q|\zeta} \right) + x_0 . \tag{23}$$

The solution for the longitudinal salinity distribution yields

$$\frac{S}{S_0} = \left\{ 1 - \frac{|Q|\zeta}{2A_0 D_{g0}} \left[ e^{(x-x_0)/\zeta} - 1 \right] \right\}^2 , \tag{24}$$

This solution is comparable to other research. It is similar with *Savenije* (2005) if $\Omega = 0$, although his solutions had an empirical Van der Burgh coefficient $K$. Besides, the solution is the same as *Kuijper and Van Rijn* (2011) if $a$ equals $b$, which implies that the depth is constant along the estuary.

With these new analytical equations, the dispersion and salinity distribution can be obtained, using the boundary conditions ($D_0$ and $S_0$).

## 3 Empirical validation and discussion

The boundary condition is often taken at the geometric inflection point ($x = x_0$) if the estuary has one. The compilation of the Muar estuary in Figure 3 is an example. Vertical dash lines display the inflection point. If there is no inflection point such as

the Landak estuary, the boundary condition is taken at the estuary mouth ($x_0 = 0$). Figure 3 demonstrates that the geometry of the alluvial estuaries fits well on a semi-logarithmic plot, supporting the exponential functions of the cross section and the width (equations (17) and (21)).

Subsequently, equation (24) is used by confronting the solution with observations, using appropriate boundary conditions. Appendix B shows how the new equation based on the maximum power concept works in twenty-three estuaries around the world. The Van der Burgh (VDB) method (*Savenije*, 2005), which has been proved to perform well in alluvial estuaries in different parts of the world and includes all mixing mechanisms, is used for comparison. Density-driven gravitational circulation is one part of the dispersive actions in estuaries. Hence, the total dispersive process from the Van der Burgh method ($D_{\mathrm{VDB}}$) must be larger than the gravitational dispersion from the maximum power method ($D_{\mathrm{MP}}$). The general geometry and measurements follow the database from *Savenije* (2012), *Gisen* (2015), and *Zhang and Savenije* (2017). The information of the VDB and MP methods is summarized in Table A. Often there is more than one salinity observation in a certain estuary (labelled by alphabet), and the observation chosen from each estuary with star-marked label is represented in Appendix B.

It can be seen that the simulated curves by the new MP method do not perform well in the wider part of the estuary (particularly upstream from the inflection point) where tidal mixing is dominant. However, the salinity observations can be very well simulated landward from the inflection point in most estuaries. In the Bernam, the Pangani, the Rembau Linggi, and the Incomati estuaries, the central part, where $D_{\mathrm{MP}}$ closely approach $D_{\mathrm{VDB}}$, is well represented. In these estuaries, the calibration is slightly lower than the observations near the intrusion limit. In general, the dispersion derived with the maximum power method declines upstream from the inflection point in agreement with the total dispersion of the empirical Van der Burgh method, which corresponds with the theory that the gravitational circulation is the dominant mixing mechanism in the landward part of these estuaries, especially in the center regime (e.g., *Hansen and Rattray*, 1965).

However, in the Thames (#8), the Delaware (#20), the Scheldt (#21), and the Pungwe (#22), the new approach seems not to work, both from the salinity and dispersion profiles. In these estuaries tide-driven mixing is dominant. Figure 4 shows the relation between the geometry and the Van der Burgh coefficient $K$ values. It can be seen that estuaries with poor performances by MP approach have lower $b/B_0$ and $K$ values. However, not all estuaries with a strongly convergent geometry perform poorly, revealing that the geometry is not the only effect. According to the expression of $\Omega$, tidal damping can play a role. In wide estuaries with strong convergence, the role of gravitational circulation is insufficient to describe the mixing. Tidal mixing processes such as lateral circulation, tidal pumping, and tidal shear are dominant. The Scheldt with preferential ebb and flood channels is a case in point (*Nguyen et al.*, 2008). Besides, the Corantijn (#9) is considered uncertain because it has a low $b/B_0$ value and contains few observations.

Overall, the maximum power approach in open systems is a useful tool to understand the mixing processes in most estuaries. In the upstream part where the effect of the tide is small, gravitational circulation plays the main role. There, the MP approach yields good results. At the same time, the predictions upstream are more relevant for water users. Where the salinity is high, it is less relevant since the water is already too saline for domestic or agricultural use.

This study provides an approach to define the dispersion coefficient due to gravitational circulation, which is proportional to the product of the dispersive velocity of the gravitational circulation and the tidal excursion length (which is the longitudinal

mixing length over which one particle travels during a tidal cycle). The dispersive velocity actually represents the strength of the density-driven mechanism. Based on the maximum power method (equation (15)), the dispersive velocity can be described as

$$v \propto \left( \frac{S|Q|T}{BE} \right)^{1/2} .$$

(25)

5    Hence, the dispersive flow due to gravitational circulation strengthens with larger freshwater discharge $|Q|$ (more stratification) and weakens with stronger tide $E$ (less stratification).

Using the calibrated dispersion coefficient at the inflection point, $C_3$ can be calculated. Except in estuaries with poor performance, $C_3$ values range from $3.5 \times 10^{-3}$ to $10.0 \times 10^{-3}$ with an average $6.8 \times 10^{-3}$ (the relative standard derivation equals 0.26). Using the average $C_3$ value to predict $D_{g0}$ (equation (18)), Figure 5 shows how the predictive equation performs. It reveals that almost all the data fall within a factor of 2, and the maximum power method underestimates the dispersion coefficient in estuaries with low $b/B_0$ values (in red) in which gravitational circulation is not enough to discribe the total dispersive processes. Finally, $R^2$ value of the regression in Figure 5 equals 0.86. Considering all the uncertainties in the measurement, $C_3$ equalling $6.8 \times 10^{-3}$ is a promising approximation to predict $D_{g0}$.

Finally, there is uncertainty about the time scale of reaching this optimum. If this time scale is longer than the tidal period, then such an optimum is not reached. In a low flow situation, however, which is the critical circumstance for salt intrusion, the variation of the river discharge is slow (following an exponential recession). If the time scale of flow recession is large compared to the time scale of salinity intrusion then it is reasonable to assume that the maximum power optimum is approached.

## 4   Combination of the MP and VDB methods

The fact that the MP method works well for density-driven mixing, but not for tide-driven mixing, whereas the VDB method works well for the combination of the two, offers an excellent opportunity for the combination of the two methods. The VDB method requires two parameters: the $K$ of Van der Burgh and the dispersion coefficient at the downstream boundary $D_0$; while the MP method only requires the downstream boundary condition $D_{g0}$. The dispersion of the VDB method, which deals with all mixing processes, should therefore always be larger than the dispersion determined by the MP method. Hence, the MP method can be used to impose an additional constraint on the calibration of the VDB method, which reduces the potential equifinality between $K$ and $D_0$. Appendix B shows the result of this mixed calibration approach: the dispersion of the VDB method is always higher than the dispersion of the MP method, and the resulting fit by the VDB method is quite acceptable.

This combined approach also allowed more accurate predictive equations as derived before. The correlation between $K$ and the estuary geometry is strong, as shown in Figure 4. This relation can be used as a predictive equation for $K$. Also the predictive equation for $D_{g0}$ is powerful, as can be seen in Figure 5, except for very wide estuaries where calibration remains necessary, and where this predictive equation can be used as a first order estimate for $D_0$.

## 5  Conclusions

An estuary is an open system which has a maximum power limit when the accelerating source is stable. This study has described a moment balance approach to non-thermal systems, yielding a new equation (15) for the dispersion coefficient due to the density-driven gravitational circulation. It shows that the dispersive action is closely related to the salinity, the freshwater discharge, the tide, and the estuarine width. This equation has been used to solve the tidal average salinity and dispersion distributions together with the steady-state equation (11). The maximum power model has then been validated with fifty salinity observations in twenty-three estuaries worldwide and compared with the Van der Burgh method. Generally, the new equation is a helpful tool to analyse the salinity distribution in alluvial estuaries, providing an alternative solution for the empirical Van der Burgh method in estuaries where the gravitational circulation is the dominant mixing mechanism. A predictive equation for dispersion at the geometric boundary has also been provided.

As can be seen in Appendix B, the gravitational dispersion is always smaller than the total effective dispersion obtained by the Van der Burgh method. In all estuaries that have a wide mouth, we see substantial tide-driven dispersion, most probably as a result of interacting preferential ebb and flood channels. This tide-driven mechanism is responsible for the (sometimes pronounced) concave slope of the salinity curve near the mouth. In the middle reach where the salinity gradient is steepest, the density-driven dispersion is all dominant and the density-driven dispersion equals the total effective dispersion. Further upstream, where the salinity gradient gradually tends to zero and the estuary becomes narrower, we see the tide-driven circulation again becoming more prominent. This is in the part of the estuary where the width to depth ratio becomes smaller and the bank shear results in more pronounced lateral velocity gradients and hence more pronounced lateral circulation. The tide-driven mixing mechanism is particularly strong in macro-tidal estuaries such as the Thames, the Scheldt, the Pungue, and the Delaware.

This study is a further development of the paper by *Zhang and Savenije* (2018), which also considered gravitational circulation based on the maximum power concept, but which did not consider the associated frictional dissipation. The approach followed in this paper maximizes the work performed by the driving gravitational torque to mix the fresh and saline water, taking account of the energy dissipation associated with this mixing. As a result, we found a solution that combines well with the empirical Van der Burgh method, providing an additional constraint for its calibration. Because the total mixing of the Van der Burgh method ($D_{\mathrm{VDB}}$) should be larger than the gravitational mixing of the maximum power concept ($D_{\mathrm{MP}}$), the calibration of the Van der Burgh method is more constrained. As a result, the Van der Burgh $K$ and the dispersion at the boundary $D_0$ can be correlated with physically observable parameters through analytical equations, which makes the Van der Burgh method a more powerful predictive model that can be applied to alluvial estuaries worldwide. More reliable observations in other estuaries are suggested to validate these maximum power and Van der Burgh methods.

*Data availability.*  About the data, all observations are available on the website at https://salinityandtides.com/.

## Appendix A: Notation

Table A1. Notations for symbols used in this study.

| Symbol | Meaning | Dimension | Symbol | Meaning | Dimension |
|---|---|---|---|---|---|
| $a$ | cross-sectional convergence length | [L] | $M$ | moment | $[ML^2T^{-2}]$ |
| $A$ | cross-sectional area | $[L^2]$ | $N_R$ | estuarine Richardson number | [-] |
| $b$ | width convergence length | [L] | $O$ | contact area | $[L^2]$ |
| $B$ | width | [L] | $P$ | Power | $[ML^2T^{-3}]$ |
| $c_s$ | saline expansivety | $[psu^{-1}]$ | $q$ | laminar resistance | $[LT^{-1}]$ |
| $D$ | dispersion coefficient | $[L^2T^{-1}]$ | $Q$ | freshwater discharge | $[L^3T^{-1}]$ |
| $D_g$ | dispersion due to gravitational circulation | $[L^2T^{-1}]$ | $S$ | salinity | [psu] |
| $E$ | tidal excursion length | [L] | $T$ | tidal period | [T] |
| $F$ | force | $[MLT^{-2}]$ | $v$ | velocity of dispersive movement | $[LT^{-1}]$ |
| $g$ | gravity acceleration | $[LT^{-2}]$ | $\delta_H$ | damping/amplifying rate | $[L^{-1}]$ |
| $h$ | depth | [L] | $\rho$ | density of water | $[ML^{-3}]$ |
| $K$ | Van der Burgh's coefficient | [-] | $\tau$ | shear stress | $[ML^{-1}T^{-2}]$ |
| $l_m$ | arm of the frictional forces | [L] | $\upsilon$ | tidal velocity amplitude | $[LT^{-1}]$ |
| $L$ | intrusion length | [L] | | | |

## Appendix B:  Application of the maximum power method

This appendix represents the application in twenty-three estuaries around the world of the maximum power method for determining the dispersion coefficient and the salinity distribution using equations (22) and (24), compared to salinity observations. The empirical Van der Burgh method is included as reference.

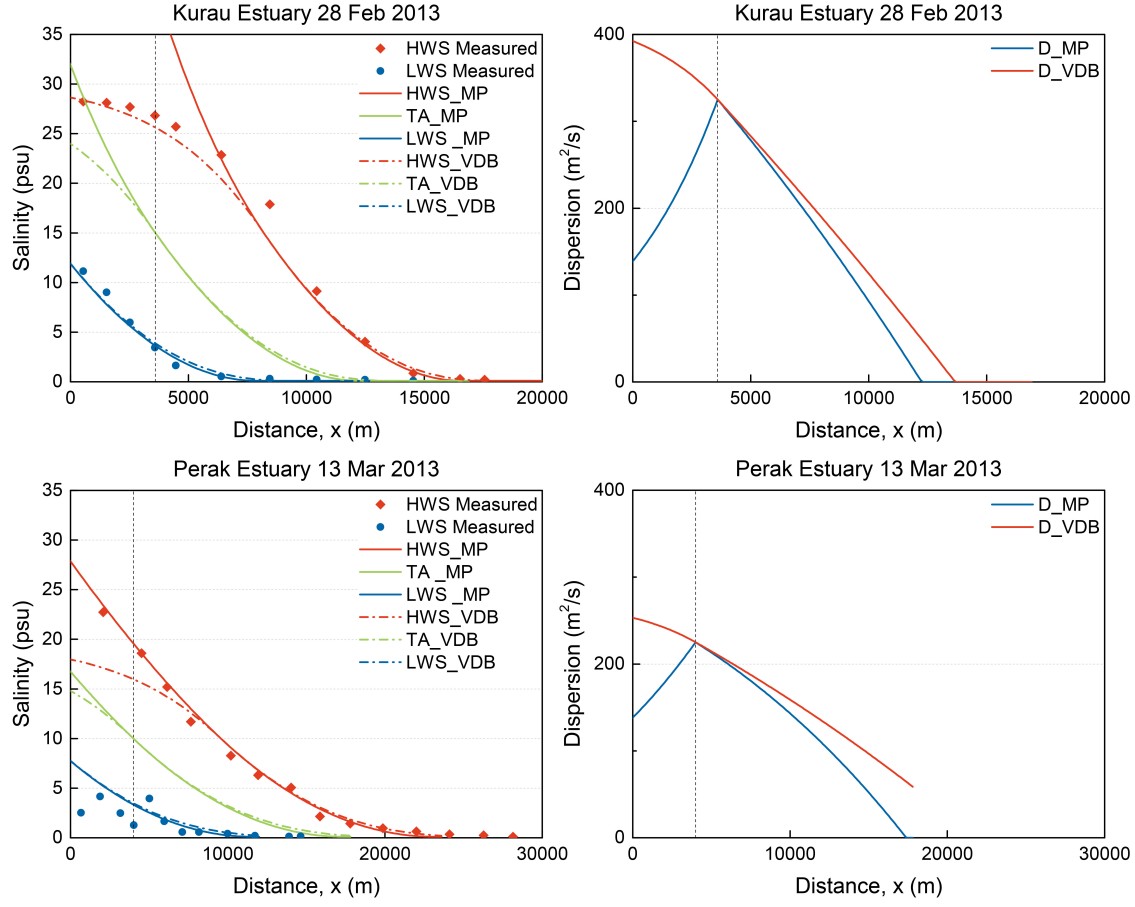

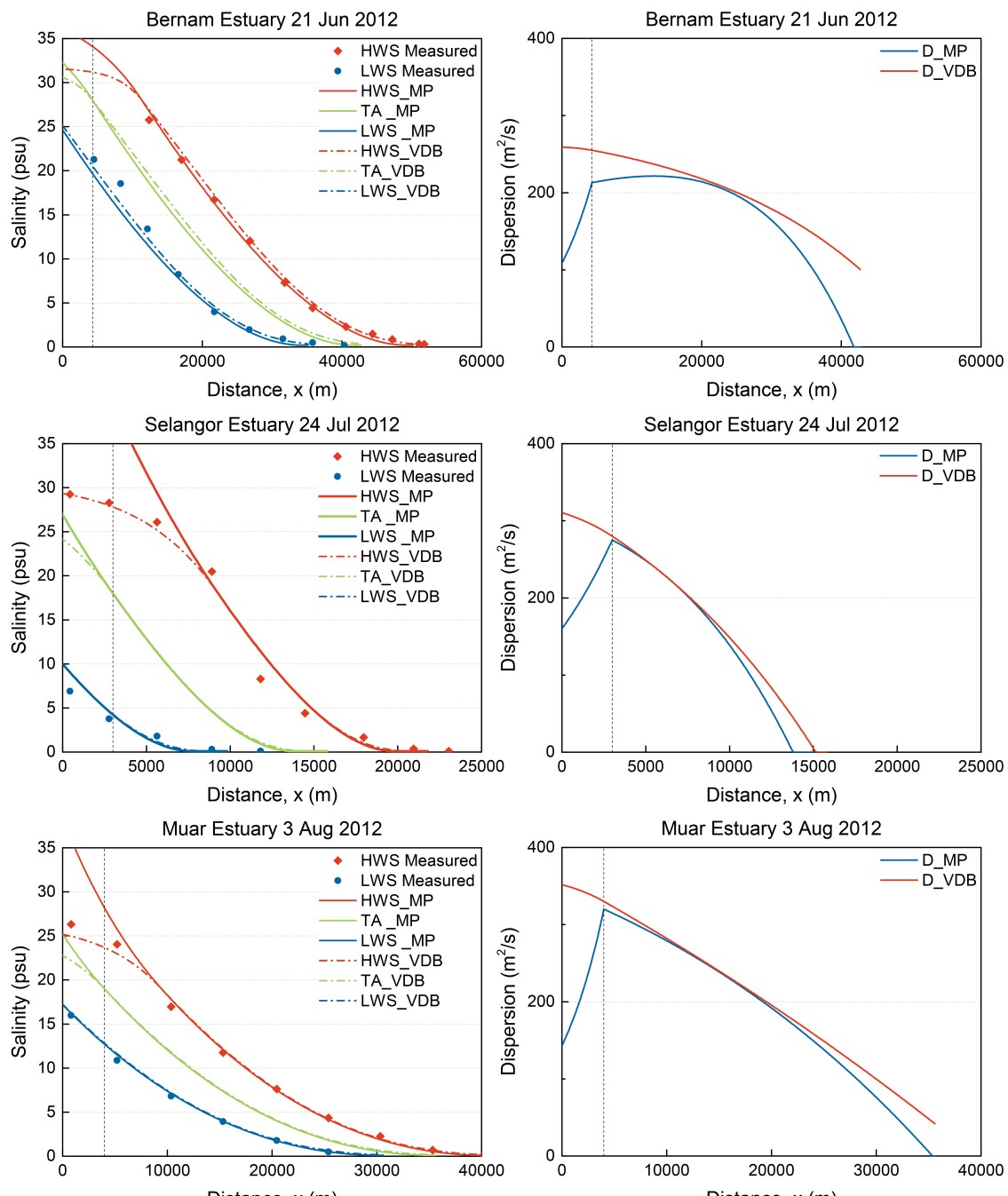

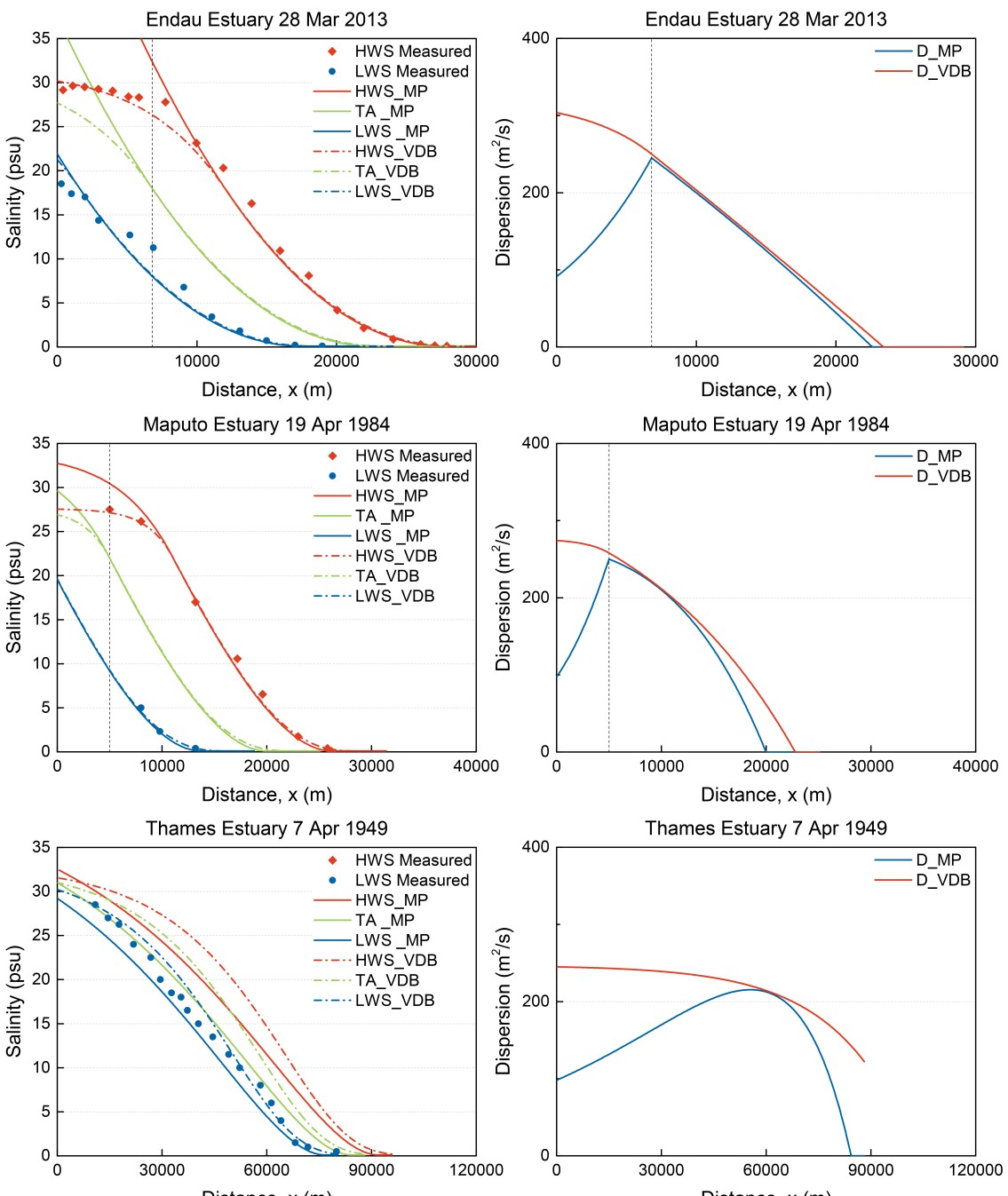

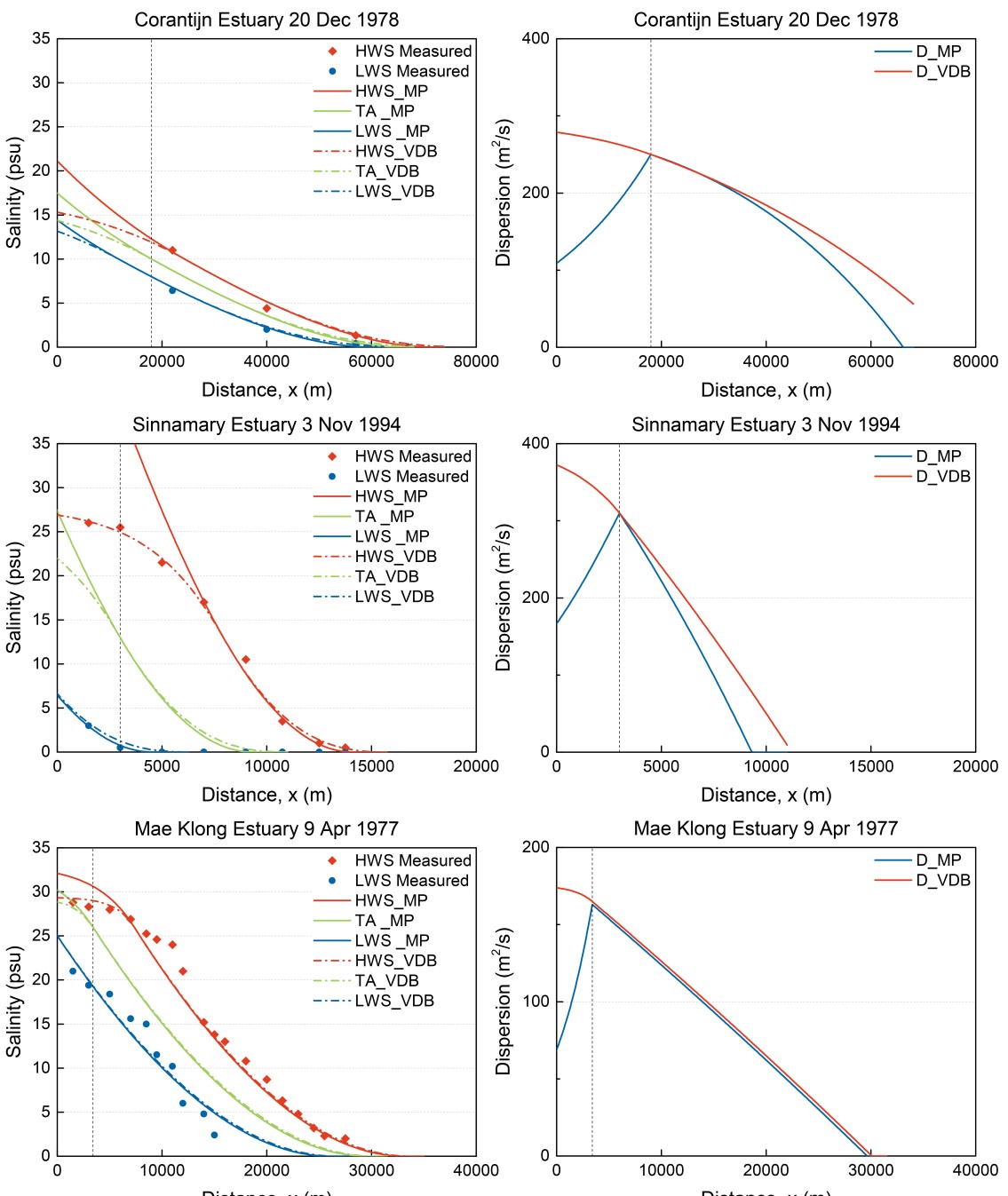

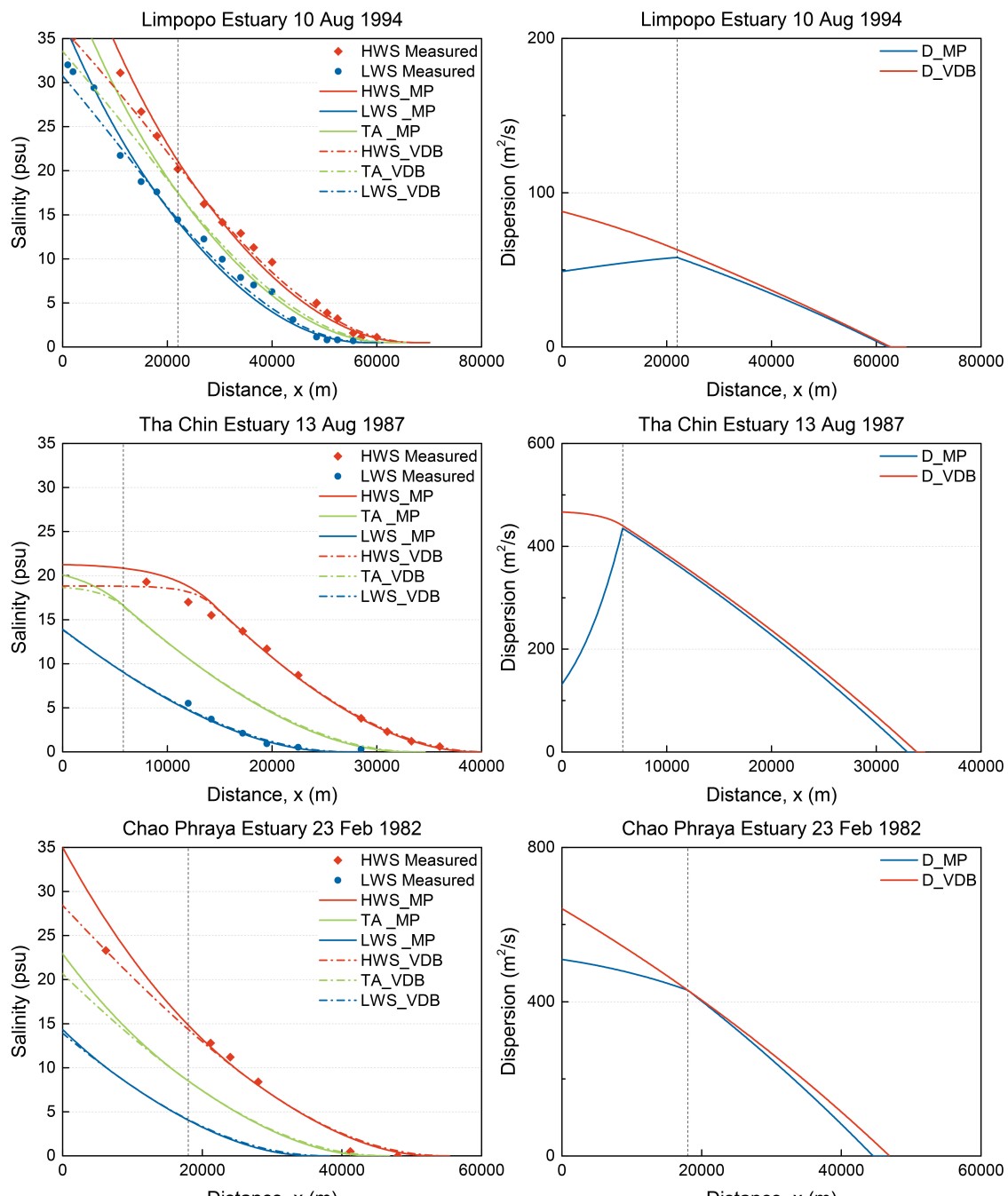

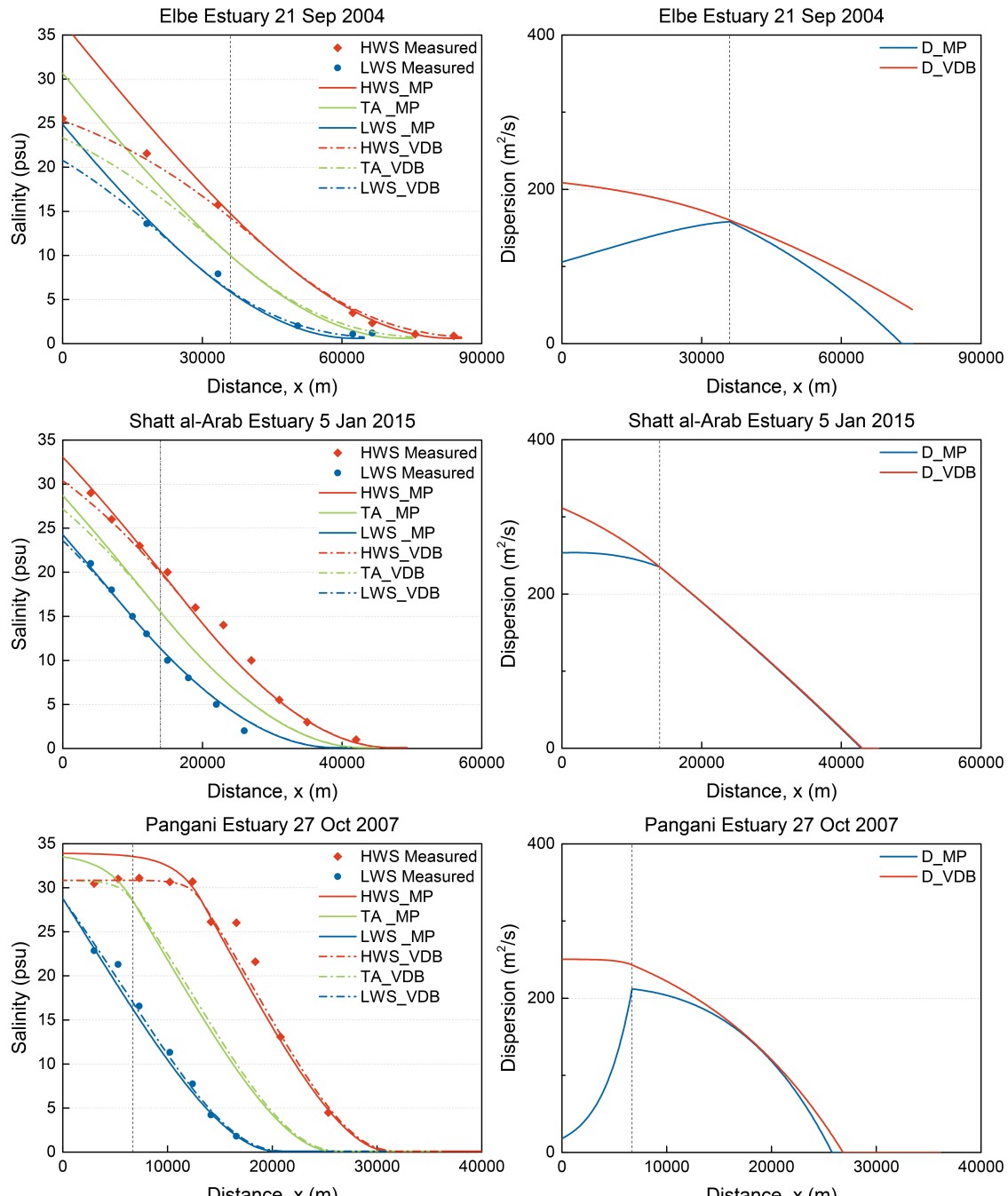

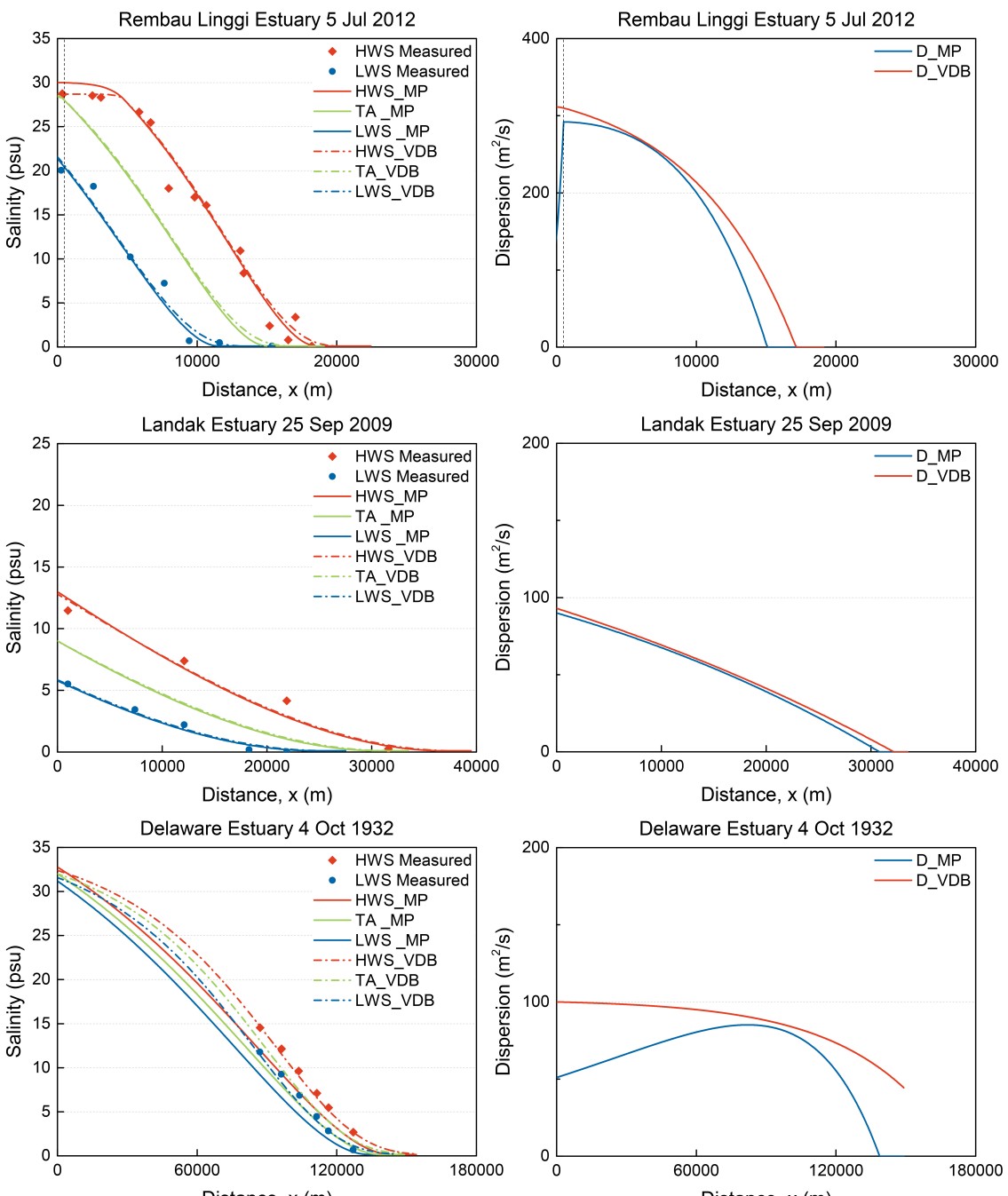

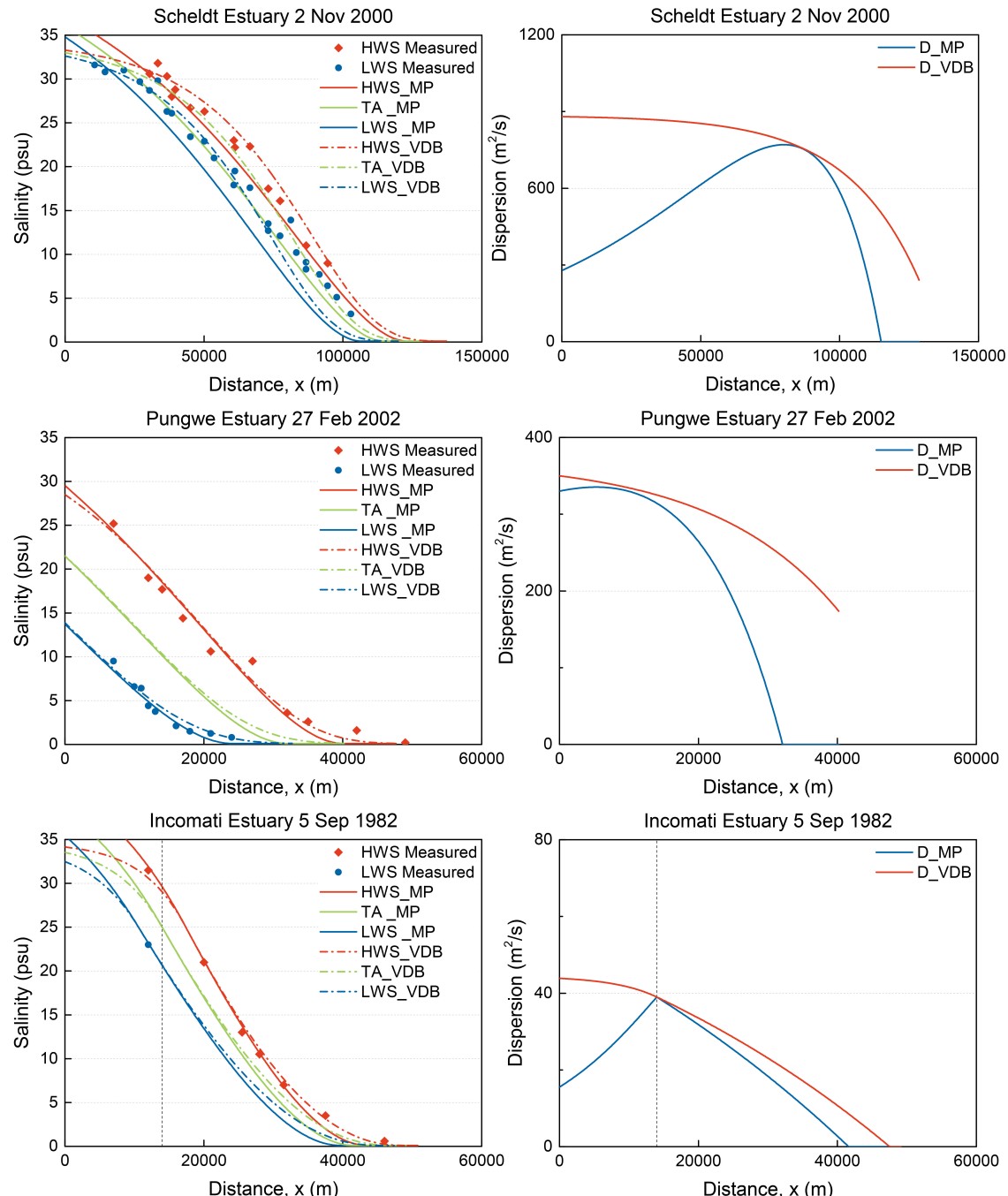

Figure A.1 Left: Application of the analytical solution from the maximum power method (solid lines) to observations (symbols) for high water slack (HWS, in red) and low water slack (LWS, in blue). The green line shows the tidal average (TA) condition. Dash dot lines reflect applications of the Van der Burgh method. Vertical dash lines display the inflection point. Right: Simulated dispersion coefficient using different methods.

*Author contributions.* Hubert Savenije conceptualised and supervised the study. Zhilin Zhang executed the research and prepared the article.

*Competing interests.* No competing interests are present.

*Acknowledgements.* The first author is financially supported for her PhD research by the China Scholarship Council.

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

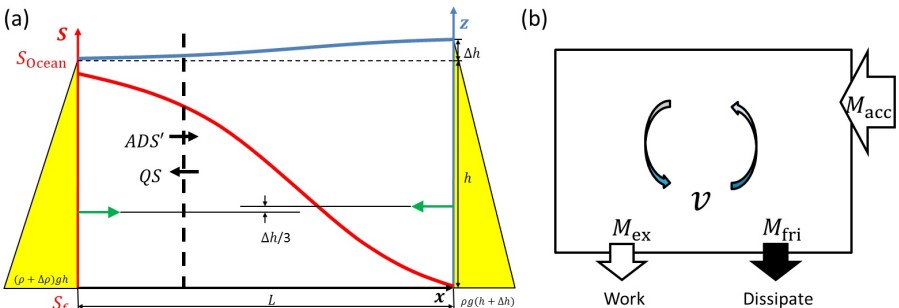

**Figure 1.** (a) Systematic salt transport in estuaries, with the seaside on the left and the riverside on the right. The water level (in blue) has a slope as a result of the salinity distributions (in red). The hydrostatic forces on both sides have different lines of action which triggers the gravitational circulation, providing an accelerating moment $M_{\mathrm{acc}}$ to the system. (b) A box-model displaying the moment balance in open estuarine systems.

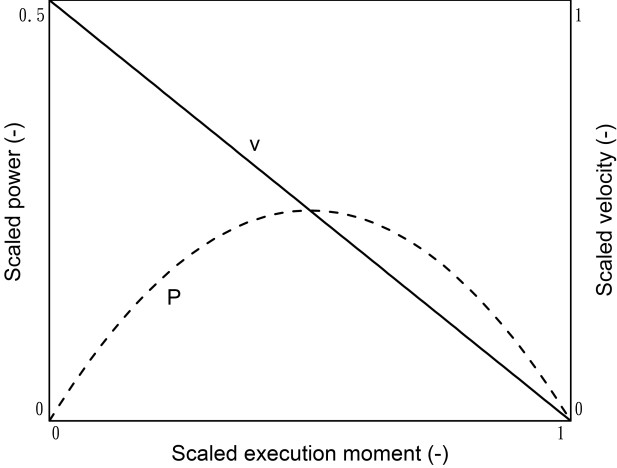

**Figure 2.** Sketch of the sensitivity of the exchange flow velocity $v$ to the working moment $M_{\mathrm{ex}}$.

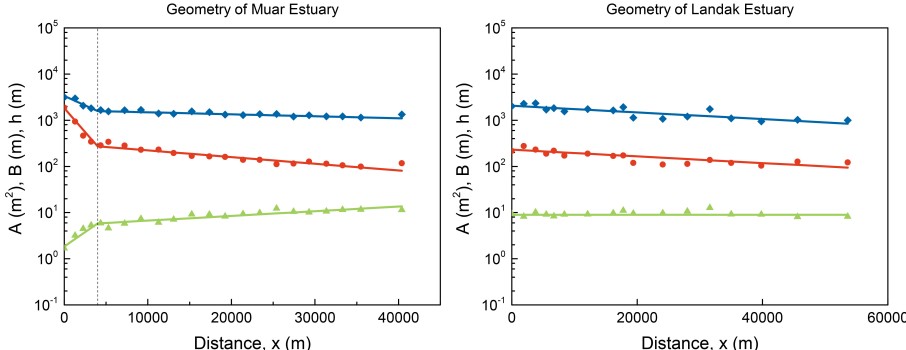

**Figure 3.** Semi-logarithmic presentation of estuary geometry, comparing simulated (lines) to the observations (symbols), including cross-sectional area (blue diamonds), width (red dots), and depth (green triangles). Vertical lines display the inflection point.

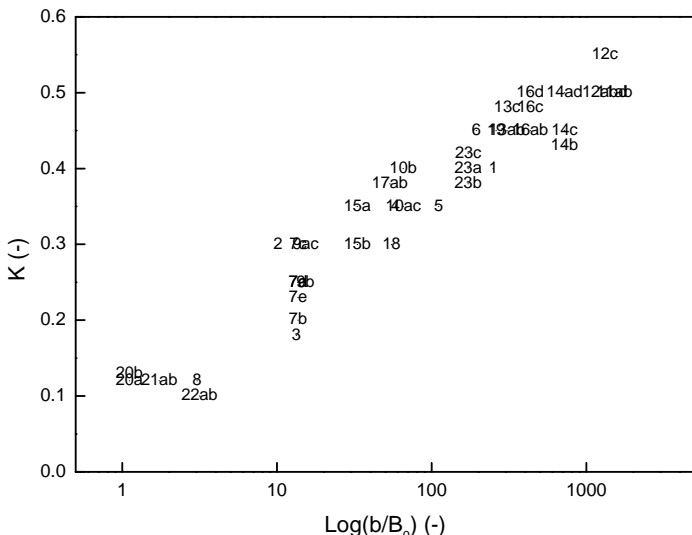

**Figure 4.** Comparison of the geometry to the Van der Burgh coefficient. Numbers show the labels of the estuaries.

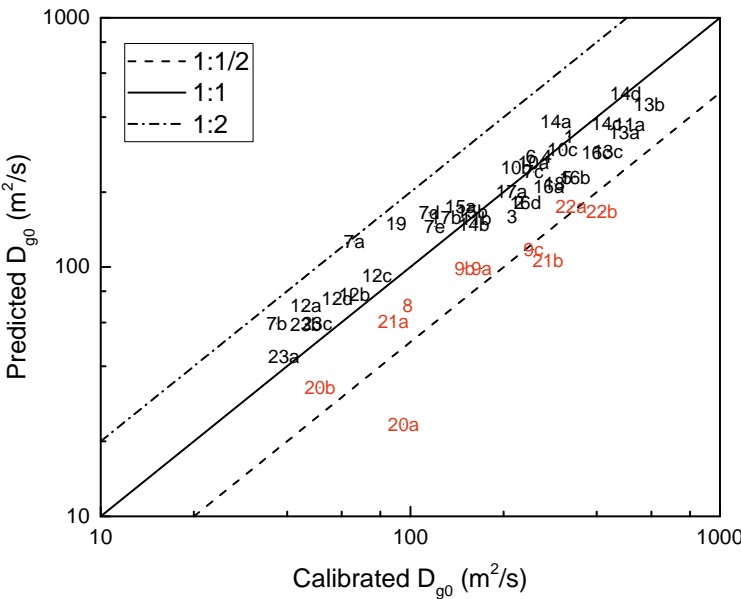

**Figure 5.** Comparison of calibrated and predicted $D_{g0}$ values by using $C_3 = 6.8 \times 10^{-3}$. Labels in red indicating the estuaries have relatively poor performance are presented for validation.

**Table 1.** Summary of application results using two methods.

| Estuary | Location | Label | $S_0$ | Maximum power | | Van der Burgh | |
|---|---|---|---|---|---|---|---|
| | | | | $D_0$ (m$^2$/s) | $C_3$ (psu$^{-1}$ms$^{-2}$) | $D_0$ (m$^2$/s) | $K$ (-) |
| Kurau | Malaysia | 1$^\star$ | 15 | 325 | 0.0064 | 325 | 0.4 |
| Perak | Mayaysia | 2$^\star$ | 10 | 225 | 0.0082 | 225 | 0.3 |
| Bernam | Malaysia | 3$^\star$ | 28 | 213 | 0.0089 | 255 | 0.18 |
| Selangor | Malaysia | 4$^\star$ | 18 | 275 | 0.0066 | 280 | 0.35 |
| Muar | Malaysia | 5$^\star$ | 19 | 320 | 0.0093 | 330 | 0.35 |
| Endau | Malaysia | 6$^\star$ | 18 | 245 | 0.0059 | 250 | 0.45 |
| Maputo | Mozambique | 7a | 29 | 66 | 0.0035 | 68 | 0.25 |
| | | 7b | 32.5 | 37 | 0.0043 | 42 | 0.2 |
| | | 7c$^\star$ | 22 | 250 | 0.0069 | 258 | 0.3 |
| | | 7d | 25 | 115 | 0.0046 | 118 | 0.25 |
| | | 7e | 26 | 120 | 0.0055 | 125 | 0.23 |
| Thames | United Kingdom | 8$^\star$ | 31 | 98 | 0.0093 | 245 | 0.12 |
| Corantijn | Suriname | 9a | 14 | 170 | 0.0114 | 170 | 0.3 |
| | | 9b | 12 | 150 | 0.0100 | 150 | 0.25 |
| | | 9c$^\star$ | 10 | 250 | 0.0141 | 250 | 0.3 |
| Sinnamary | French Guiana | 10a | 8 | 250 | 0.0063 | 250 | 0.35 |
| | | 10b | 6.5 | 220 | 0.0058 | 220 | 0.4 |
| | | 10c$^\star$ | 13 | 310 | 0.0070 | 310 | 0.35 |
| Mae Klong | Thailand | 11a | 24 | 510 | 0.0090 | 520 | 0.5 |
| | | 11b$^\star$ | 26 | 163 | 0.0069 | 165 | 0.5 |
| Limpopo | Mozambique | 12a | 23 | 46 | 0.0044 | 51 | 0.5 |
| | | 12b | 13 | 66 | 0.0056 | 70 | 0.5 |
| | | 12c | 16 | 78 | 0.0056 | 92 | 0.55 |
| | | 12d$^\star$ | 17.5 | 58 | 0.0051 | 63 | 0.5 |
| Tha Chin | Thailand | 13a | 23 | 490 | 0.0094 | 490 | 0.45 |
| | | 13b | 25.5 | 590 | 0.0087 | 600 | 0.45 |
| | | 13c$^\star$ | 16.5 | 435 | 0.0099 | 440 | 0.48 |
| Chao Phraya | Thailand | 14a | 11 | 295 | 0.0051 | 305 | 0.5 |
| | | 14b | 1 | 160 | 0.0071 | 165 | 0.43 |
| | | 14c$^\star$ | 8.5 | 430 | 0.0076 | 430 | 0.45 |

| Estuary | Location | Label | $S_0$ | Maximum power | | Van der Burgh | |
|---------|----------|-------|-------|---------------|---|---------------|---|
| | | | | $D_0$ (m$^2$/s) | $C_3$ (psu$^{-1}$ms$^{-2}$) | $D_0$ (m$^2$/s) | $K$ (-) |
| | | 14d | 12 | 495 | 0.0066 | 510 | 0.5 |
| Elbe | Germany | 15a | 10 | 145 | 0.0055 | 150 | 0.35 |
| | | 15b$^\star$ | 10 | 158 | 0.0063 | 160 | 0.3 |
| Shatt al-Arab | Iraq | 16a | 11.5 | 280 | 0.0088 | 280 | 0.45 |
| | | 16b | 16 | 340 | 0.0099 | 340 | 0.45 |
| | | 16c | 27 | 400 | 0.0092 | 400 | 0.48 |
| | | 16d$^\star$ | 15.5 | 235 | 0.0086 | 235 | 0.5 |
| Pangani | Tanzania | 17a$^\star$ | 28.5 | 212 | 0.0070 | 243 | 0.38 |
| | | 17b | 28 | 130 | 0.0054 | 145 | 0.38 |
| Rembau Linggi | Malaysia | 18$^\star$ | 28 | 292 | 0.0090 | 310 | 0.3 |
| Landak | Indonesia | 19$^\star$ | 9 | 90 | 0.0040 | 93 | 0.45 |
| Delaware | United States | 20a | 11 | 95 | 0.0269 | 200 | 0.12 |
| | | 20b$^\star$ | 32 | 51 | 0.0103 | 100 | 0.13 |
| Scheldt | Netherlands | 21a | 31 | 88 | 0.0097 | 225 | 0.12 |
| | | 21b$^\star$ | 33 | 278 | 0.0173 | 800 | 0.12 |
| Pungwe | Mozambique | 22a$^\star$ | 21.5 | 330 | 0.0124 | 350 | 0.1 |
| | | 22b | 20 | 415 | 0.0165 | 500 | 0.1 |
| Incomati | Mozambique | 23a$^\star$ | 25 | 39 | 0.0058 | 39 | 0.4 |
| | | 23b | 17 | 46 | 0.0052 | 46 | 0.38 |
| | | 23c | 16 | 50 | 0.0056 | 50 | 0.42 |