# Peer review of "MAXIMUM POWER OF SALINE AND FRESH WATER MIXING IN ESTUARIES"

_Earth System Dynamics, 2018_

## Short Comment (SC1) · 27 Dec 2018

1. The paper presents interesting view for saline intrusion in estuaries. It deserves a publication after some revisions. 2. For salt intrusion in estuaries, the driving agents are diverse, rather than the gravitational circulation only. As mentioned in the manuscript, tidal mixing is another key element. Even in the upstream part of an estuary, the gravitational circulation is still not the main driver, other processes like tidal pumping, tidal trapping are more important. How to ensure the validity of your concept? 3. I have not got any expression of the Mex in your manuscript. How the Mex is expressed and how it can reach a maximum? 4. The dispersion of salinity is rather a very ambiguous term, what is its physical meaning? how to define and describe it?

---

## Author Comment (AC1) · 14 Jan 2019

1. The paper presents interesting view for saline intrusion in estuaries. It deserves a publication after some revisions.

Reply: Thank you for your positive comment.

2. For salt intrusion in estuaries, the driving agents are diverse, rather than the gravitational circulation only. As mentioned in the manuscript, tidal mixing is another key element. Even in the upstream part of an estuary, the gravitational circulation is still not the main driver, other processes like tidal pumping, tidal trapping are more important. How to ensure the validity of your concept?

Reply: Indeed, the tide generally plays a dominant role downstream, particularly if

the estuary is wide. As shown in Appendix B, D_MP is always smaller than D_VDB, which is the total dispersion we simulated by the traditional Van der Burgh method and which comes close to the dispersion distribution that can be derived from the observed salinity distribution and the salt balance equation. The difference between the D_MP and the D_VDB represents the tide driven dispersion.

Over a tidal cycle, tidal trapping is minor in well-defined single-channel estuaries. Tidal pumping is apparent in preferential ebb and flood channels (occurring in wide estuaries) and by the interaction of the tidal flow with an irregular bathymetry, which both can be important near the estuarine mouth.

However, at the location where the salinity gradient is largest, the dominant mechanism is the gravitational circulation.

3. I have not got any expression of the Mex in your manuscript. How the Mex is expressed and how it can reach a maximum?

Reply: As you have noticed, our approach to analyse the energy dissipation in estuarine systems is different from traditional research. The accelerating moment exercised by the freshwater discharge transfers energy either into dissipation by friction or into work by the executing moment, M_ex, which drives the gravitational circulation, lifting up saline water against gravity and pushing down fresh water against gravity. We did not know the exact expression for M_ex, but at maximum power, half of the accelerating moment is converted into work: the executing moment.

The maximum power happens when dP/dM_ex = 0, this is because the power is a function of the execution moment (Figure 2) and there is a maximum value of the quadratic Power function.

4. The dispersion of salinity is rather a very ambiguous term, what is its physical meaning? how to define and describe it?

Reply: Indeed, the dispersion coefficient is a mathematical artifact of averaging. Depending on the spatial and temporal scale of averaging, it arrives at different values. In this manuscript, we consider the tidal average and cross-sectional average dispersion along the estuary. The nice thing is that if we use the tidal period and tidal excursion length as the temporal and spatial scale, that the dispersion gets physical meaning: it represents the mixing (the exchange) of fresh and saline water over a tidal cycle within a cross-section. A box model demonstrates that this dispersion coefficient then is the product between the tidal excursion and the velocity of salinity exchange between two sections a tidal excursion apart. The latter is supposed to be proportional to the tidal velocity amplitude and to be a function of the degree of stratification.

---

## Referee Comment (RC1) · Anonymous Referee #1 · 2 Apr 2019

This papers presents an elegant, analytical model to describe salt intrusion in estuarine systems. Symbols / abbreviations are consistently used while accounting for the correct units. In its compactness, the paper tends to be a bit brief on providing background information / references on distinct aspects and explaining the rationale behind certain choices. For instance: • The fundamental principle underlying the new model ('freely evolving systems perform work and dissipate energy at maximum power, close to the Carnot limit') is only briefly introduced; a more extensive description of this concept, preferably illustrated with one or two examples would be helpful. • The estuarine geometry used in the model (Eq. 16, 17, 21). Why these expressions? Do we know from earlier studies that these fit well with estuary geometries across the world? Reference? • The geometric inflection point of an estuary (Section 3). How

is this defined? Reference to literature?

On the set-up of the model: • Based on the description of gravitational circulation and the definition sketch in Fig. 1, I would expect the horizontal length scale of the circulation to relate to the length of the salt wedge (= distance L in Fig. 1) rather than the tidal excursion E (which is the distance the salt wedge travels up and down the estuary between high tide and low tide). Please clarify. If so, does it affect the model formulations? • The model does not cater for a bed slope along the estuary. How would inclusion of such bed slope, even if minor, affect the gravitational circulation (order of magnitude analysis)? If of secondary importance, please state.

On the model outcomes / presentation of results: The presented figures clear show the model potential to represent the salinity dispersion across the majority of estuaries considered. Nevertheless some questions remain: • From the presented results, it is not clear which estuary corresponds to the numbers listed in Figures 4 and 5 • Why does the MP method calculate an (erroneous) strong decrease of salinity values seaward of the inflection point? If not realistic, isn't it better to leave this part of the model output out? • How much parameter fittings is needed to achieve the results presented here? Is it only the C3 value, or are other parameters modified as well? Were the geometry parameters varied as part of the calibration? • The estuaries labelled in red show larger deviations than the other ones. This becomes clear from Fig. 5 (not from Fig 4 yet – though indicated there as 'less reliable datasets'). What can be the physical explanation for this? In what sense are the red estuaries different from the other ones? Please clarify further on the explanation of model devations. • Calibrated and predicted values of DgO differ on a log-log scale. What is the implication of this in terms of deviations in calculated salinity profile? In other words, how sensitive is the model to offsets in C3. After having gone through this paper, the reader may wonder about the added value of this new model – as the existing Van der Burgh method generally gives better results (especially seaward of the infliction point). It would be good to clearly stipulate the benefits and added value of the new model in

the paper, to avoid any possible confusion at this point.

---

## Referee Comment (RC2) · Axel Kleidon (Referee) · 4 Apr 2019

This manuscript describes the application of the maximum power limit to the mixing in estuaries. The work is based on an earlier publication of the same authors and extends it to a more refined explanation. I found the manuscript novel, innovative, and it was generally well written. I think the manuscript needs mostly technical corrections so I recommend a minor revision. In addition, I think the authors miss an opportunity by not describing the relationship to their previous work in greater detail and draw some conclusions from what can be learned. I think this can provide insights beyond estuaries on the application of optimality principles, and this would fit nicely to the scope of the special issue.

General comments:

**1 Relation to previous work. I think the manuscript would benefit if the relationship to the authors' earlier work is more clearly described and discussed. This concerns the introduction and the discussion/conclusion. It would really help the reader to understand if the previous work contained errors or whether it was an approximation? I find the current description about the previous work was limited by using an isolated systems' view. This is difficult to understand for a reader that is not completely familiar with the earlier work, so this needs a more detailed description and explanation.**

**2 Terminology. In the manuscript, the term "moment" is used. Do the authors mean momentum? Angular momentum? Torque? This is not clear to me (I think you mean torque), so I think it would be helpful to briefly describe/clarify it at the beginning.**

Minor comments:

- Abstract: I found it not so easy in the abstract to distinguish between background knowledge and the contribution by this paper. A sentence somewhere with "Here we show" or similar would help to clarify this distinction.

- Page 1, Line 18: What is a "working line"?

- Page 2, Line 13: What do you mean by "accelerating energy"? And is Nfric not the friction force, rather than energy dissipation, which should be the product of Nfric and the velocity?

- Page 2, Line 15: I would clarify it here that you write an angular momentum balance here.

- Page 4, Line 4: I am not an expert in estuaries. Is the one-dimensional advection-dispersion equation standard knowledge? If so, it would be useful to add a standard reference here.

- Page 6: In the evaluation section, I found that I missed some information. Where

does the data come from? Also, an overview, like a table, of the different estuaries and where they are located would be helpful.

- Page 6, Line 8: "paper"? Do you mean a semi-logarithmic plot?

- Page 6, Line 13: I would it helpful to know more about the Van der Burgh method so I can understand better what is being compared. Specifically, what are the main differences of the VDB method compared to maximum power? This does not need to be extensive, but a brief summary of how the VDB method works would be helpful. At this point, the parameter of the Van der Burgh method, K, should also be introduced and described. Also, how does K relate to the parameter C3? They are compared in Table 1, but at present, I do not know what this comparison means. Are they supposed to be the same?

- Page 6, Line 16: The abbreviation MP has not been defined.

- Page 6, Line 17: The reference to the Table is broken.

- Page 6, Line 37: "too saline to use" - to use for what?

- Page 8, Lines 8-10. As mentioned above, I think there is more that can be learned here by comparing this work to the previous work of the authors. At the moment, this is rather short. I think the authors miss an opportunity here to contrast this approach to the previous one. This should help to identify what one can learn in terms of system setup when applying optimality approaches. I think such a more extended discussion would be very suitable to the context of the special issue on optimality principles.

- Page 8, Table A1: "regularly" — better "in this study".

- Page 9: Please explain the terms used in the legend, such as HWS, LWS etc. Also, it would help to relate the estuaries to the ones listed in Table 1.

- Page 23, Figure 3: Please describe what the vertical lines are in the caption.

- Page 23, Figure 4: Please explain what the different symbols are, on the axes and in

the Figure (link it to the estuaries in Table 1).

- Page 25, Table 1: What does "label" refer to? What is "S0"? What does the "*" refer to in the lines? This table needs more description. Also, I think it refers to the different estuaries, so it would really help to add the names of the estuaries here as well.

Disclosure:

I also want to mention here that I know the second author, Hubert Savenije, very well, and served on the PhD committee of the first author, Zhilin Zhang. I do not think, however, that this impacts my judgement of this work.

---

## Author Comment (AC2) · 25 Apr 2019

We would like to thank referee #1 for the discussion.

1. The fundamental principle underlying the new model ('freely evolving systems perform work and dissipate energy at maximum power, close to the Carnot limit') is only briefly introduced; a more extensive description of this concept, preferably illustrated with one or two examples would be helpful.

Reply: This manuscript is closely connected and a follow-up research of the previous paper by Zhang and Savenije (2018). The maximum power concept is well introduced in this paper. We shall summarize it in the revised version to facilitate the reader.

2. The estuarine geometry used in the model (Eq. 16, 17, 21). Why these expressions?

[Figure]

Do we know from earlier studies that these fit well with estuary geometries across the world? Reference?

Reply: The compilation of geometry using exponential equations is well documented by, such as in: Savenije (2005, 2012, 2015), Gisen (2015), and Zhang and Savenije (2018).

3. The geometric inflection point of an estuary (Section 3). How is this defined? Reference to literature?

Reply: The geometric inflection point is well defined and described by Savenije (2005, 2012, 2015). On the seaside of this point, the morphology of the estuary mouth is dominated by wave energy, beyond this point the morphology is dominated by the kinetic energy of the tide.

4. Based on the description of gravitational circulation and the definition sketch in Fig. 1, I would expect the horizontal length scale of the circulation to relate to the length of the salt wedge (= distance L in Fig. 1) rather than the tidal excursion E (which is the distance the salt wedge travels up and down the estuary between high tide and low tide). Please clarify. If so, does it affect the model formulations?

Reply: The tidal excursion is the distance that a water particle travels up and down the estuary during a tidal cycle. As a result, the tidal excursion is the length scale of the mixing process. The water particles do not travel the entire salt intrusion length during a tidal cycle, but circulate back and forth over the tidal excursion. All particles in the salt intrusion length L perform gravitational circulation within a distance E.

5. The model does not cater for a bed slope along the estuary. How would inclusion of such bed slope, even if minor, affect the gravitational circulation (order of magnitude analysis)? If of secondary importance, please state.

Reply: In alluvial estuaries, the bottom slope is small compared to the ratio of the increase of depth due to the salinity difference ($\Delta h$) to the salt intrusion length (L). Moreover, it does not affect the residual water slope $\Delta h/L$, which results from the difference in hydraulic pressure, which is independent on the bottom slope. If, a downward slope were introduced in the picture, then we would have to include the horizontal component of the bottom pressure as well: the water pressure over the additional depth near the downstream boundary would then be balanced by the horizontal component of the sea water pressure near the bottom of the estuary mouth. This would make the sketch unnecessary complex.

6. From the presented results, it is not clear which estuary corresponds to the numbers listed in Figures 4 and 5.

Reply: Numbers are the labels of estuaries in Table 1. Two columns name the estuaries and their locations have been added in Table 1 in Page 25-26. We shall make this clear in the caption of the revised paper.

7. Why does the MP method calculate an (erroneous) strong decrease of salinity values seaward of the inflection point? If not realistic, isn't it better to leave this part of the model output out?

Reply: Near the estuarine mouth, the width is large and the convergence length is small, and the dispersion by gravitational circulation $D\_g$ is small according to Equation (15) in Page 4. In this case, $D\_g$ is by far not enough to describe the saline and fresh water mixing and, as a result, it bends down the salinity curve. We leave this strong decrease to show how the salinity would look like if there was no tidal mixing and all mixing would be density driven.

8. How much parameter fittings is needed to achieve the results presented here? Is it only the C3 value, or are other parameters modified as well? Were the geometry parameters varied as part of the calibration?

Reply: For the dispersion coefficient by gravitational circulation, $D\_g\_0$ (or $C\_3$ using Equation (18) in Page 5) is the only parameter to be calibrated. Besides $D\_g\_0$, the

geometry is the determining factor but this is given for each estuary.

9. The estuaries labelled in red show larger deviations than the other ones. This becomes clear from Fig. 5 (not from Fig 4 yet – though indicated there as 'less reliable datasets'). What can be the physical explanation for this? In what sense are the red estuaries different from the other ones? Please clarify further on the explanation of model deviations.

Reply: Figure 4 has been edited; all labels are black now. In Figure 5, labels in red indicate the estuaries have relatively poor performance, as described in Lines 25-34, Page 6.

10. Calibrated and predicted values of Dg0 differ on a log-log scale. What is the implication of this in terms of deviations in calculated salinity profile? In other words, how sensitive is the model to offsets in C3.

Reply: In this log-log scale figure, it is easier to show the relationship between the calibrated and calculated values even in the small value range. In a linear plot, the lower values would plot close to the line of perfect agreement. In this research, we have not considered the sensitivity to $C_3$.

11.After having gone through this paper, the reader may wonder about the added value of this new model – as the existing Van der Burgh method generally gives better results (especially seaward of the infliction point). It would be good to clearly stipulate the benefits and added value of the new model in the paper, to avoid any possible confusion at this point.

Reply: First of all, this new model provides a physical explanation for the good performance of the Van der Burgh model in the region where gravitational circulation is the dominant mixing process. The Van der Burgh model, surprisingly enough, also works well in the part where tidal mixing is dominant. The very simple Van der Burgh model thus has a wider empirical applicability. The practical importance of the new model is,

that it provides an additional constraint to the calibration of the Van der Burgh model. The Van der Burgh model has two degrees of freedom (the calibrated K and D_0). This new model, following the maximum power concept, has only one parameter to be determined. The fact that the dispersion by the gravitational circulation (of the maximum power model) should be smaller than the dispersion of all mixing mechanisms combined (of the Van der Burgh model) provides an additional constraint on the Van der Burgh method, which has two parameters to be calibrated (the Van der Burgh coefficient K and D_0), which may partly compensate each other. With this restriction, the Van der Burgh method is more accurate and more powerful.

Additional References: Savenije, H. H. G.: Prediction in ungauged estuaries: An integrated theory, Water Resour. Res., 51, 2464–2476, 2015.

---

## Author Comment (AC3) · 25 Apr 2019

We would like to thank referee #2 for the discussion.

**1 Relation to previous work. I think the manuscript would benefit if the relationship to the authors' earlier work is more clearly described and discussed. This concerns the introduction and the discussion/conclusion. It would really help the reader to understand if the previous work contained errors or whether it was an approximation? I find the current description about the previous work was limited by using an isolated systems' view. This is difficult to understand for a reader that is not completely familiar with the earlier work, so this needs a more detailed description and explanation.**

Reply: We shall add a more detailed description of the deficiencies in the earlier work

and how this paper carries the concept of maximum power further. The following additional description shall be added at the end of Page 1: 'In this research, the maximum power concept for estuarine mixing is further elaborated. The equation derived earlier in Zhang and Savenije (2018) appeared to have an analytical solution of a straight line for the salinity distribution. With hindsight this is reasonable: In the center region, where the salinity gradient is at its maximum, dominated by density-driven mixing, the salinity decreases linearly in the upstream direction (Hansen and Rattray, 1965). It can be seen as a first order solution. However, this approximate solution was not fully satisfactory for simulating the salinity distribution. When choosing the boundary conditions at the correct system boundaries, the derivation appears to work better.'

**2 Terminology. In the manuscript, the term "moment" is used. Do the authors mean momentum? Angular momentum? Torque? This is not clear to me (I think you mean torque), so I think it would be helpful to briefly describe/clarify it at the beginning.**

Reply: It is indeed the torque. '(torque)' has been added in Line 13, Page 2 after 'an angular moment'.

Minor comments: -Abstract: I found it not so easy in the abstract to distinguish between background knowledge and the contribution by this paper. A sentence somewhere with "Here we show" or similar would help to clarify this distinction.

Reply: 'Accordingly', in Line 9, Page 1, has been changed to 'In this research'. Then the following content is the outcome of the research.

- Page 1, Line 18: What is a "working line"?

Reply: Working line is the line of action, which is a geometric representation of how the force is applied. It is the line through the point at which the force is applied in the same direction as the vector force. 'working lines' in the manuscript has been changed into 'lines of action' in this article.

-Page 2, Line 13: What do you mean by "accelerating energy"? And is Nfric not the

friction force, rather than energy dissipation, which should be the product of Nfric and the velocity?

Reply: Accelerating energy is the energy source from fresh water discharge. N_fric is the energy dissipation due to friction F_fric.

-Page 2, Line 15: I would clarify it here that you write an angular momentum balance here.

Reply: It is a moment balance equation, which has a dimension of [ML^2T^{-2}].

-Page 4, Line 4: I am not an expert in estuaries. Is the one-dimensional advection-dispersion equation standard knowledge? If so, it would be useful to add a standard reference here.

Reply: This equation comes from the combination of water balance equation and salt balance equation, which is well described by Savenije (2005, 2012). The reference has been added.

- Page 6: In the evaluation section, I found that I missed some information. Where does the data come from? Also, an overview, like a table, of the different estuaries and where they are located would be helpful.

Reply: In Line 15, Page 6, 'The general geometry and measurements follow the database from Savenije (2015), Gisen (2015), and Zhang and Savenije (2017)'. In addition, numbers are the labels of estuaries in Table 1. Two columns name the estuaries and locations have been added in Table 1 in Page 25-26.

- Page 6, Line 8: "paper"? Do you mean a semi-logarithmic plot?

Reply: It has been edited.

- Page 6, Line 13: I would it helpful to know more about the Van der Burgh method so I can understand better what is being compared. Specifically, what are the main differences of the VDB method compared to maximum power? This does not need to

be extensive, but a brief summary of how the VDB method works would be helpful. At this point, the parameter of the Van der Burgh method, K, should also be introduced and described. Also, how does K relate to the parameter C3? They are compared in Table 1, but at present, I do not know what this comparison means. Are they supposed to be the same?

Reply: The Van der Burgh method is a well-performing empirical model to describe the salinity intrusion in estuaries, which considers all the mixing mechanisms mainly including the gravitational circulation and tidal effects. Its application requires two parameters to be calibrated (the empirical Van der Burgh coefficient K and the dispersion coefficient at the boundary condition D_0). The maximum power method, however, considers only the gravitational circulation due to freshwater discharge, which has only one parameter D_g_0 to calibrate. The dispersion distribution D(x) obtained by the maximum power method (based on gravitation mixing only) must be smaller than that of the Van der Burgh method, which provides an additional constraint on the Van der Burgh method. According to our research, K is primarily connected to the estuarine geometry and to a lesser extent to the ratio of the fresh and saline water flows into the estuary. In addition, C_3 has the same value for almost all the estuaries. In Table 1, we merely present the values and we do not compare K and C_3. The following paragraph has been added at the end of the Introduction Section: 'The Van der Burgh (VDB) method is a well-documented approach to solve the salinity intrusion in estuaries (Savenije, 2005, 2012), which takes account of all mixing mechanisms, including the density-driven gravitational circulation and tide-driven mechanisms. For the application of this method, there are two parameters that need to be calibrated, the empirical Van der Burgh coefficient K and the dispersion coefficient at the downstream boundary D_0. This method has performed surprisingly well around the world and has been as the benchmark model for the maximum power model in this research.'

- Page 6, Line 16: The abbreviation MP has not been defined.

Reply:'MP' has been introduced in Lin 14, Page 3.

- Page 6, Line 17: The reference to the Table is broken.

Reply: It has been edited.

- Page 6, Line 37: "too saline to use" - to use for what?

Reply: Too saline for domestic use.

- Page 8, Lines 8-10. As mentioned above, I think there is more that can be learned here by comparing this work to the previous work of the authors. At the moment, this is rather short. I think the authors miss an opportunity here to contrast this approach to the previous one. This should help to identify what one can learn in terms of system setup when applying optimality approaches. I think such a more extended discussion would be very suitable to the context of the special issue on optimality principles.

Reply: Indeed, it is significant to refer to previous work. The last paragraph of the manuscript in Page 8 has been added: 'This study is a further development of the paper by Zhang and Savenije (2018), which also considered gravitational circulation based on the maximum power concept, but using a different system boundary. The approach followed in this paper uses a broader system boundary and found a solution that combines well with the empirical Van der Burgh method, providing an additional constraint for its calibration. Because the total mixing of the Van der Burgh method (D_VDB) should be larger than the gravitational mixing of the maximum power concept (D_MP), the calibration of the Van der Burgh method is more constrained. As a result, the Van der Burgh K and the dispersion at the boundary D_0 can be correlated with physically observable parameters through analytical equations, which makes the Van der Burgh method a predictive model that can be applied to alluvial estuaries world-wide.'

- Page 8, Table A1: "regularly" âAËŸT better "in this study". ËĞ

Reply: It has been edited.

- Page 9: Please explain the terms used in the legend, such as HWS, LWS etc. Also,

it would help to relate the estuaries to the ones listed in Table 1.

Reply: The information has been added. HWS represents high water slack; LWS represents low water slack; TA represents the tidal average condition. Two columns name the estuaries and locations have been added in Table 1 in Page 25-26.

- Page 23, Figure 3: Please describe what the vertical lines are in the caption.

Reply: The vertical lines represent the location of the geometric inflection point. It has been mentioned in Line 6, Page 6, which is now added in the caption of Figure 3.

- Page 23, Figure 4: Please explain what the different symbols are, on the axes and in the Figure (link it to the estuaries in Table 1).

Reply: Indeed, the numbers represent the estuaries. Table 1 has been edited.

- Page 25, Table 1: What does "label" refer to? What is "S0"? What does the "*" refer to in the lines? This table needs more description. Also, I think it refers to the different estuaries, so it would really help to add the names of the estuaries here as well.

Reply: The labels were used to distinguish the estuaries. $S_0$ is the salinity at the inflection point, which has been mentioned in Line 4, Page 5. There may be more than one observations in a certain estuary, and '*' is used to show which observations were chosen to be shown in Appendix B. The observations including the labels and estuaries' name follows the dataset in the previous article (Zhang and Savenije, 2018), but as mentioned above, two columns name the estuaries and locations have been added in Table 1 in Page 25-26.

Additional Reference: Hansen, D. V. and Rattray, M.: Gravitational circulation in straits and estuaries, J. Mar. Res., 23, 104–122, 1965. Zhang, Z.: A theoretical basis for salinity intrusion in estuaries, Ph.D. thesis, Delft University of Technology, 2019.